# Effect of Nanostructured Scaffold on Human Adipose-Derived Stem Cells: Outcome of *In Vitro* Experiments

**DOI:** 10.3390/nano10091822

**Published:** 2020-09-12

**Authors:** Marina Borgese, Ludovica Barone, Federica Rossi, Mario Raspanti, Roberto Papait, Luigi Valdatta, Giovanni Bernardini, Rosalba Gornati

**Affiliations:** 1Department of Biotechnology and Life Sciences, University of Insubria, via J.H. Dunant 3, 21100 Varese, Italy; m.borgese@uninsubria.it (M.B.); lbarone1@studenti.uninsubria.it (L.B.); federica.rossi@uninsubria.it (F.R.); roberto.papait@uninsubria.it (R.P.); luigi.valdatta@uninsubria.it (L.V.); giovanni.bernardini@uninsubria.it (G.B.); 2Department of Medicine and Surgery, University of Insubria, Via Guicciardini 9, 21100 Varese, Italy; mario.raspanti@uninsubria.it

**Keywords:** human Adipose derived Stem Cells, regenerative medicine, biomaterial, extracellular matrix

## Abstract

This work is addressed to provide, by *in vitro* experiments, results on the repercussion that a nanostructured scaffold could have on viability, differentiation and secretion of bioactive factors of human adipose-derived stem cells (hASCs) when used in association to promote angiogenesis, a crucial condition to favour tissue regeneration. To achieve this aim, we evaluated cell viability and morphology by MTT (3-[4,5-dimethylthiazol-2-yl]-2,5-diphenyltetrazolium bromide) assay and microscopy analysis, respectively. We also investigated the expression of some of those genes involved in angiogenesis and differentiation processes utilizing quantitative polymerase chain reaction (qPCR), whereas the amounts of Vascular Endothelial Growth Factor A, Interleukin 6 and Fatty Acid-Binding Protein 4 secreted in the culture medium, were quantified by enzyme-linked immunosorbent assay (ELISA). Results suggested that, in the presence of the scaffold, cell proliferation and the exocytosis of factors involved in the angiogenesis process are reduced; by contrast, the expression of those genes involved in hASC differentiation appeared enhanced. To guarantee cell survival, the construct dimensions are, generally, smaller than clinically required. Furthermore, being the paracrine event the primary mechanism exerting the beneficial effects on injured tissues, the use of conditioned culture medium instead of cells may be convenient.

## 1. Introduction

Cell biological processes are regulated by a multitude of extracellular signals, and recent studies suggest that interactions between extracellular matrix (ECM) components with cell surface receptors play an important role [1]. The ECM is a network of extracellular molecules that affect matrix characteristics such as geometry, elasticity, physical and mechanical signals, and topographical features involved in mechanotransduction, a process by which the cells translate mechanical stimuli into biochemical signals [2,3,4]. Several reports demonstrated that natural and synthetic 3D nanostructured matrices interact with various surface receptors and molecules, triggering specific cell responses such as proliferation, migration, morphogenetic movements and cell fate, including stem cell lineage commitment [5,6,7,8]. It has been also reported that cell adhesion to the ECM contributes to angiogenesis and epithelial development [9,10]. Nanotopography refers to specific surface features at the nanoscopic scale (from 1 to 100 nm) and this term is commonly used to define nanotextured surfaces of biomaterials [5]. Appropriate nanotopography is required for physiological regeneration and embryogenesis of the stem cell niche; nevertheless, the mechanism behind topography-induced cellular response is still unclear [2,11]. Furthermore, recent findings indicate the implication of topographical cues of new biomaterials also in modulating paracrine functions of human mesenchymal stem cells (MSCs) [12]. Among the different stem cell populations, human adipose-derived stem cells (hASCs), a subpopulation of MSCs, due to their self-renewal, multi-lineage differentiation capacity, but particularly for their ability to release biologically active molecules able to induce angiogenesis [13,14], are very interesting in many medical applications, such as regenerative medicine.

Unfortunately, it has been observed that in *in vivo* implantation of cell-loaded scaffolds, only the cells within a distance of 100–200 μm to the nearest capillaries receive sufficient oxygen and nutrients to maintain their metabolism and function; these issues restrict construct dimensions to smaller than clinically required, limiting the use of cell-loaded scaffolds for *in vivo* applications due to the inability to adequately vascularize the compromised tissues [15].

In this scenario, it is no wonder that research is directed towards investigating the interactions between cells and the biomaterials to comprehend whether it is possible, for *in vivo* practice, to replace cells with cellular derivatives, such as the conditioned medium, able to maintain the same angiogenic potential.

Here, we report the outcomes of *in vitro* experiments in which hASCs have been grown in the presence of Flowable Wound Matrix (FWM), an advanced wound care device used to treat difficult access wounds and tunnelled wounds.

The results shown in this paper intend to provide new information to the body of knowledge about the potential of cell-free devices in the recovery of damaged tissues.

## 2. Materials and Methods

### 2.1. Flowable Wound Matrix

The Flowable Wound Matrix (Integra^®^) was kindly provided and characterized by LifeSciences Corporation (Plainsboro, NJ, USA; www.integralife.com). FWM is an advanced wound care device comprised of granulated cross-linked bovine tendon collagen and glycosaminoglycan. The granular collagen is hydrated with saline and applied in difficult-to-access wound sites and tunnelled wounds.

### 2.2. Human Adipose-Derived Stem Cell (hASC) Isolation and Culture

The hASCs were obtained from mammary adipose tissue of three healthy women who underwent mammoplasty reduction procedure. The subjects gave their informed consent for inclusion in the study, and all the procedures were performed in accordance with the Ospedale di Circolo (Varese, Italy) Ethical Committee and the European Communities Council Directive of EU/63/2010 (24 April 2013, n° 302). The average age of the subjects was 43 ± 4 years. They were non-smokers, had no history of metabolic disorders, were not taking medications at the time of the medical procedure, and had not experienced any great weight loss from dieting (body mass index (BMI) was from 18.8 to 29 kg/m^2^). Owing to the mammoplasty reduction procedure, we were able to obtain large amounts of adipose tissue from the same depot; furthermore, the rigorous selection of the subjects insured that specimens were of good quality and rich in hASCs.

The hASCs were isolated according to the Gronthos and Zannettino protocol modified in our laboratory [16]. Briefly, the stromal vascular fraction (SVF) was obtained by collagenase type II digestion (Sigma Aldrich, Milano, Italy) at 37 °C for 1 h in agitation. SVF was filtered (100 μm cell strainers) and centrifuged at 180× *g* for 10 min; the resulting pellet was washed with erythrocyte lysis buffer (154 mM NH_4_Cl, 10 mM KHCO_3_, and 1 mM EDTA (Ethylenediaminetetraacetic acid)), then seeded in T25 flasks maintained at 37 °C, 5% CO_2_. After 6 h, non-attached cells were removed. Cells were grown in DMEM:DMEM/F12 1:1 (Sigma Aldrich, Milano, Italy), supplemented with 2 mM L-Gln, 1% penicillin-streptomycin, 0.1% gentamicin, and 10% FBS (Fetal Bovine Serum). hASCs were subsequently cultured in T75 flasks and used at the 5th confluence for all experiments.

### 2.3. hASC Characterization

Cells were characterized by cytofluorimetric analysis, immunostaining and quantitative polymerase chain reaction (qPCR); the methods and the results were described in Cherubino [16] and Borgese [17].

For cytofluorimetric analysis, we used a series of monoclonal antibodies (mAb) specific for staminal markers CD44, CD90, CD105, for the differentiation marker CD45 and for major histocompatibility molecules including HLA class I (HLA-A, B, C) and class II (HLA-DR). For immunostaining, CD44 antibody was used as a stemness marker and adiponectin receptor 1 (ADIPOR1) antibody as an adipogenic differentiation marker.

For qPCR, *CD44* and *CD90* genes were used as positive stemness markers, while *fatty acid-binding protein 4* (*FABP4*), *adipocyte complement-related protein 30* (*ACRP30*), and *acetyl-coenzyme A synthetase 2* (*ACSS2*) genes were taken as differentiation markers. According to the method of Palombella [18] *glyceraldehyde-3-phosphate dehydrogenase*-*GAPDH* and *beta2-microglobulin*-*β2m*) were used as reference genes. Quantification has been conducted by using the 2^−ΔΔCt^ method.

### 2.4. Flowable Wound Matrix (FWM)-hASC Mixture Preparation

FWM was supplied sterile, in single use kits containing one syringe with granular bovine tendon collagen and glycosaminoglycan, one empty sterile syringe, a luer lock connector and one flexible injector. To conduct the *in vitro* experiments, Integra^®^ FWM was rehydrated with 3 mL of culture medium containing the appropriate cell number, and then mixed according to the manufacturer’s instructions (Figure 1). The scaffold could be considered ready for use when the product appeared uniform and homogeneous, assuming a gel-like consistency.

### 2.5. Cell Viability/Proliferation Assay

A cell proliferation assay was carried out by an MTT (3-[4,5-dimethylthiazol-2-yl]-2,5-diphenyltetrazolium bromide) absorbance-based method. About 800 μL of rehydrated FWM, containing 10,000 cells in culture medium, were placed in a 24-well plate. To ensure a uniform cell distribution in the scaffold, the plate was shaken for 1 min, and then grown for 24, 48, 72 or 96 h. In another set of experiments the same number of cells were seeded without FWM.

The MTT assay (Cayman Chemicals, Milan, Italy) is based on the enzymatic reduction, by mitochondrial succinate dehydrogenase, of the water soluble MTT to an insoluble formazan. The formazan is then solubilized, and the concentration determined by optical density at 570 nm using the Infinite F200 plate reader (Tecan Group, Männedorf, Switzerland). The experiments were performed on 3 subjects and run in triplicate; values were expressed as mean + standard error (S.E.).

### 2.6. Optical and Electron Microscopy

For microscopy, about 800 μL of rehydrated FWM, containing 10,000 cells, were placed in a 24-well plate and grown for 24, 48, 72 or 96 h, and then processed as reported in Pirrone [19]. Briefly, samples were fixed in Karnovsky solution (2% glutaraldehyde, 4% formaldehyde in 0.1 M sodium cacodylate buffer, pH 7.4) for 24 h at 4 °C and then preserved in 0.1 M sodium cacodylate buffer at 4 °C. Specimens were post-fixed using 1% OsO_4_ in 0.1 M sodium cacodylate buffer (pH 7.4), dehydrated in ethanol (50%, 70%, 90% and 100%) and embedded in Epon-Araldite 812 as described in Bava [20].

For histological analysis, 7 μm-thick sections were cut (MR3 Motorized Microtome, RMC Boeckeler), stained with hematoxylin and eosin, mounted using Eukitt (Bio-Optica, Milan, Italy) and then observed using a Zeiss Axiovert200 optical microscope (Zeiss, Milan, Italy) equipped with a Charge-Coupled Device (CCD) camera TrueChrome HD IIS (TiEsseLab, Milan, Italy).

For transmission electron microscopy (TEM), 90 nm-thick sections were cut using a Pabisch Top-Ultra A ultra-microtome (Emme 3 Srl, Lainate, Italy), collected on Cu/Rh 300 mesh grids, counterstained with uranyl acetate and lead citrate, and examined using a JEOL-1010 electron microscope (JEOL, Tokyo, Japan) operating at 90 kV and equipped with a MORADA CCD camera (Olympus, Tokyo, Japan).

For scanning electron microscopy (SEM), 50,000 cells, in presence of FWM, were cultured on a 24 mm^2^ coverslip, placed into a six-well culture plates, and maintained for 24 h. Samples were washed with phosphate-buffered saline (PBS), fixed in Karnovsky solution, dehydrated and treated with hexamethyldisilazane (HMDS). Samples were mounted on standard SEM stubs with conductive carbon-based adhesive, and gold-coated in an Emitech K-550 sputter-coater (Emitech Ltd., Ashford, UK) in a controlled argon atmosphere at a pressure of 0.1 mbar. All observations were carried out on a FEI XL-30 FEG SEM field-emission scanning electron microscope (FEI, Eindhoven, Netherlands) operated at an acceleration voltage of 7 kV.

### 2.7. Gene Expression

We placed 800 μL of rehydrated FWM, containing about 10,000 cells in culture medium, in a 24-well plate and grown for 24, 48, 72 and 96 h. In another set of experiments, the same number of cells were seeded without FWM.

Total RNA was isolated using the Direct-zol RNA Miniprep (Zymo Research, Milan, Italy) according to manufacturer’s instructions. The extracted RNA was quantified by the QuantiFluor^®^ RNA System (Promega, Milan, Italy) and assessed by 1% gel electrophoresis to verify the integrity. The RNA was reverse transcribed using the iScript™ cDNA Synthesis Kit (BioRad, Segrate, Italy) and the cDNA was stored at −20 °C until use. The qPCR was performed with iTaq™ Universal SYBR^®^ Green Supermix technology (Bio-Rad, Segrate, Italy) using a CFX Connect^®^ Real-Time PCR Detection System apparatus (Bio-Rad, Segrate, Italy). The Beacon Designer 7^®^ Program (Bio-Rad, Segrate, Italy) was used to design specific primers. In order to prevent genomic DNA amplification, the primers were designed on an exon-exon junction (possibly separated by an intron of at least 1000 bp). The sequences are shown in Table 1. Each sample, prepared as reported in Rossi [21], consist of 1 µL (5 ng) of cDNA, 1 µL of forward and reverse primer mix (6 µM), 7.5 µL of SYBR Green Supermix (2x), and water to a final volume of 15 µL were mixed and placed in the CFX 96 Thermocycler (BioRad, Segrate, Italy). The thermal cycle comprised the following settings: 5 min at 95 °C, 10 s at 95 °C, and 30 s at 60 °C for 40 cycles. The analysis was performed on genes involved in angiogenesis such as *Vascular Endothelial Growth Factor A* (*VEGF-A*), *Interleukin 6* (*IL6*) and *Hypoxia Inducible Factor 1* (*HIF1α*), and genes indicative of adipogenic differentiation such as *Fatty Acid Binding Protein 4* (*FABP4*), *Acetyl-Coenzyme A Synthetase 2* (*ACSS2*), and *Adipocyte Complement-Related Protein 30* (*ACRP30*). Values, obtained from 3 subjects, were normalized as described in the paragraph “hASC characterization”. Each experiment was repeated three times.

### 2.8. Enzyme-Linked Immunosorbent Assay (ELISA) on hASC Culture Medium

To evaluate the proteins secreted by hASCs, enzyme-linked immunosorbent assay (ELISA) was performed following the manufacturer’s instructions (BioVendor, Brno, Czech Republic); 48 h after sowing, in presence or not of FWM, the hASCs were maintained for 48 additional hours in serum-free culture, and then the assay was performed on VEGFA, IL6, and FABP4. The protein amount was determined acquiring the absorbance at 450 nm using the Infinite F200 plate reader (Tecan Group, Männedorf, Switzerland) of three different subjects. Each experiment was repeated 3 times and values were expressed as mean ± S.E.

### 2.9. Statistical Analysis

Statistical analysis, on values obtained by MTT assay, was conducted according to Bonferroni’s test. *** *p* ≤ 0.001 (*n* = 3).

## 3. Results

### 3.1. Cell Viability

In order to evaluate the impact that the collagen scaffold had on the hASCs, we first evaluated the viability of these cells grown in FWM. The MTT assay showed that hASCs, grown in the absence of FWM, proliferated normally reaching confluence after 96 h of incubation. Conversely, the FWM caused a decrease of cell viability of 20%, 30% and 40% at respectively 48, 72 and 96 h, after sowing of hASCs (see Figure 2).

### 3.2. Microscopy Observation

Using optical microscopy (Figure 3), we confirmed the presence of the hASCs inside the scaffold, even though we also observed the presence of some apoptotic cells (see the inset of Figure 3).

We then assessed the impact of FWM on cell morphology and ultrastructure (Figure 4). TEM observation revealed that in presence of the scaffold, hASCs are induced to form structures resembling pseudopods (see Figure 4A); furthermore, SEM confirmed cell colonization, represented by round or ovoid cells inside the scaffold (Figure 4B).

### 3.3. Gene Expression

We also analysed the impact of FWM on the ability of hASCs to promote angiogenesis rather than differentiate into adipocytes (see Figure 5). In order to achieve this aim, we evaluated the expression of some genes that play a key role in these two biological processes. qPCR showed that the mRNA content of pro-angiogenic genes (*IL-6, VEGFA* and *HIF1α*) appeared to be down-regulated (Figure 5A); whereas, genes related to adipogenesis (*ACRP30, ACSS2* and *FABP4*), were upregulated in cells grown in presence of the scaffold (Figure 5B).

### 3.4. Enzyme-Linked Immunosorbent Assay (ELISA)

The loss of the ability of hASCs, grown in FWM, to release angiogenic growth factors was confirmed by measurement of the amount VEGFA and IL-6 secreted in the culture medium (Figure 6). ELISA, performed on medium isolated from hASCs grown in FWM for 24, 48, 72 and 96 h, showed a decrease in the levels of these growth factors compared to the cells grown in the absence of the scaffold (Figure 6A,B). The content of FABP4, a protein primarily expressed in adipocytes, is appreciable only after 96 h in the culture medium of hASCs cultivated in FWM (Figure 6C).

## 4. Discussion

Adult human tissues are capable of self-renewal as they have a reserve of staminal cells and, recently, it has been demonstrated that tissue regeneration and angiogenesis take place preferably by the secretion of trophic factors rather than directly by differentiation of stem cells [12,22]. MSCs, particularly hASCs, produce and secrete a broad variety of cytokines, chemokines, and growth factors, whose characteristics may depend on the composition of ECM. As the mechanisms behind the interactions between ECM and cellular response are still unclear, when a scaffold is used in association with hASCs an important aspect to take into account is how it can influence cell processes.

In this paper we tested the effects that FWM, a collagen scaffold, had on the hASCs. This scaffold is widely used to treat difficult access wounds and tunnelled wounds, as it provides an optimal milieu for cellular invasion and capillary growth.

Notwithstanding the above, our data on viability highlighted that, *in vitro*, FWM hampers the proliferation of hASCs, and this is probably due to time-limited access to nutrients, CO_2_ removal, and lack of stiffness of the scaffold. In fact, in our previous work [16], we demonstrated that when a bilayer matrix, composed of a collagen sheet with glycosaminoglycans and a silicone layer, able to provide an adherent and rigid covering, is used, hASCs seeded on it can survive and proliferate at the same rate as those grown in the absence of the scaffold. In this regard, although the stiffness optima for different kinds of adherent cells vary widely [23], Haugh [24] reported results on the effect on the compressive modulus of collagen glycosaminoglycan scaffolds on seeded cells, suggesting that the stiffest scaffolds induce an increase of cell number. However, in the body tissue, it is known that mammalian cells, requiring nutrients, oxygen and waste exchanges for their survival, are located within 100–200 μm of blood vessels [25]. In our experiments, we used 800 µl of scaffold mixture, and it was no surprise that the cells located at the centre of the scaffold, being over 200 µm from the interface with the culture medium, died. Despite the lack of proliferation, the surviving cells undergo a cytoskeletal organization forming structures resembling pseudopods that allow scaffold colonization; this result is also an indication of a good cytocompatibility of the FWM. This finding is in accordance with data from literature describing the influence of collagen-glycosaminoglycan scaffold on actin cytoskeleton [5,26,27]. We also found that the collagen scaffold promotes the differentiation of hASCs into adipocytes, as shown by gene expression analysis of several markers of adipocytes differentiation. These results are in agreement with data from literature that show that cell–matrix interactions, involving the actin cytoskeleton, can lead to cell differentiation [26,27], and scaffold architecture is capable of directing lineage specification. To support this assumption, it has been reported that human MSCs and the induced pluripotent stem cells, cultured on different composites, i.e., polycarbonate membrane and collagen sponges or polylactide-co-glycolide scaffolds impregnated with fibronectin and type I collagen, as well as magnesium-enriched hydroxyapatite, show an increase of the osteogenic potential, in some cases, mediated by MAPK and PI3 kinase pathways [28,29,30,31,32]. Furthermore, Crespi et al. [33], in an *in vivo* study on 15 healthy patients, reported the outcome of maxillary sinus grafting with autologous bone to that obtained with magnesium-enriched hydroxyapatite (mHA). mHA graft showed significantly higher expression of the osteoblast differentiation factors and the bone matrix formation markers compared to the autologous bone group.

All this evidence may explain and support the results we got on mRNA level demonstrating that the hASCs, grown in the presence of FWM for 96 h, expressed higher *ACRP30, ACSS2* and *FABP4* mRNA compared to those maintained in the absence of the scaffold.

ACRP30, also referred to as adiponectin, is the most abundant adipocytokine secreted by adipocytes and plays a pivotal role in glucose metabolism and energy homeostasis [34,35]. Although ACRP30 influences the proliferation of hASCs, it is also linked to adipocyte and osteoblast differentiation: an increase of *ACRP30* mRNA expression and protein secretion have been observed in hASCs during their differentiation into osteoblast [36,37,38].

Acyl CoA synthetase, or ACSS2, is a cytosolic enzyme, well conserved from bacteria to humans, that catalyses the activation of acetate for its use in lipid synthesis and energy generation [39,40]. Recently it has been shown that *ACSS2* gene expression increased under hypoxia in tumour cells [41,42]; thus, the increase of this gene expression that we found in hASCs grown in the scaffold could be consequence of a limited oxygenation of the scaffold.

Although *FABP4*, commonly known as adipocyte protein 2, has been extensively used as a marker for differentiated adipocytes, it is little expressed also in stem cells [43,44]. It seems that exogenous FABP4, acting on hASCs regulates about 64 key node proteins, most of which are kinases, affecting the process of cell differentiation rather than cell proliferation and growth [45]. Finally, we provide evidence that the ability of hASCs to secrete some angiogenic factors (IL-6, VEGFA and HIF1α) is compromised by a collagen scaffold. Growth factors are intercellular signalling molecules, subjected to an intricate cellular pathway, and able to regulate a variety of cellular processes such as migration, proliferation, and differentiation in a mode that depends on the cell type and microenvironment conditions [46,47]. In this scenario, it is clear that in medical applications, when the angiogenesis is the primary process to be promoted, it is necessary to guarantee the presence of specific growth factors in the microenvironment. In view of the results obtained, suggesting that the presence of the scaffold promotes the differentiation of hASCs to the detriment of their angiogenetic potential, and considering that it is the paracrine signalling the primary mechanism that exert the beneficial effects on injured tissues [15,25], for regenerative medicine and tissue engineering, in the foreseeable future, it is reasonable to consider the concept of a cell-free therapy avoiding invasive techniques in favour of mininvasive ones [48,49].

These data provide a solid, rational approach for the use of the conditioned culture medium as a promising therapeutic treatment in the field of regenerative medicine and tissue engineering. In this regard, preliminary *in vivo* experiments, conducted in our laboratory, on athymic BALB/c nude mice grafted with four different conditions (1-FWM alone; 2-FWM associated with hASCs; 3-FWM associated with hASC-crude protein extract; 4-FWM associated with hASC-conditioned medium) sustained our assumption.

Furthermore, in our laboratory, we are currently conducting extracellular vesicle separation, from hASC-conditioned medium, followed by their purification and characterization. Although the discovery of extracellular vesicles goes back more than 30 years, they have garnered recent interest due to their potential diagnostic and therapeutic relevance. In fact, the use of a combination of complementary growth factors, such as those present in stem cell-conditioned medium, could be more advantageous compared to the addition of single growth factors or, even better, stem cell transplantation. Furthermore, the absence of an immune response to the cell-conditioned medium can bypass some issues and difficulties associated with cell transplant.

## Figures and Tables

**Figure 1 nanomaterials-10-01822-f001:**
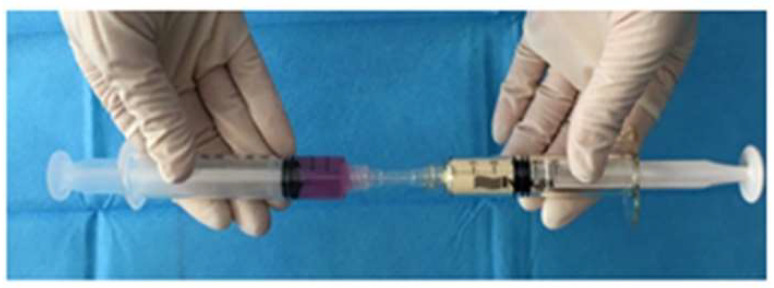
Flowable Wound Matrix-human adipose-derived stem cell (FWM-hASC) mixture preparation.

**Figure 2 nanomaterials-10-01822-f002:**
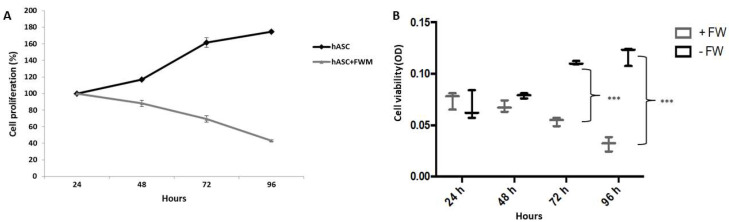
MTT viability assay. Regression (**panel A**) and Box plot (**panel B**) of hASC viability maintained in presence or not of FWM. hASCs were grown in the presence (grey line) or absence (dark line) of FWM at different time points (24, 48, 72 and 96 h) after plating. FWM results in a decrease of cell viability of hASCs. Statistical analysis was conducted according to Bonferroni’s test. *** *p* ≤ 0.001 (*n* = 3).

**Figure 3 nanomaterials-10-01822-f003:**
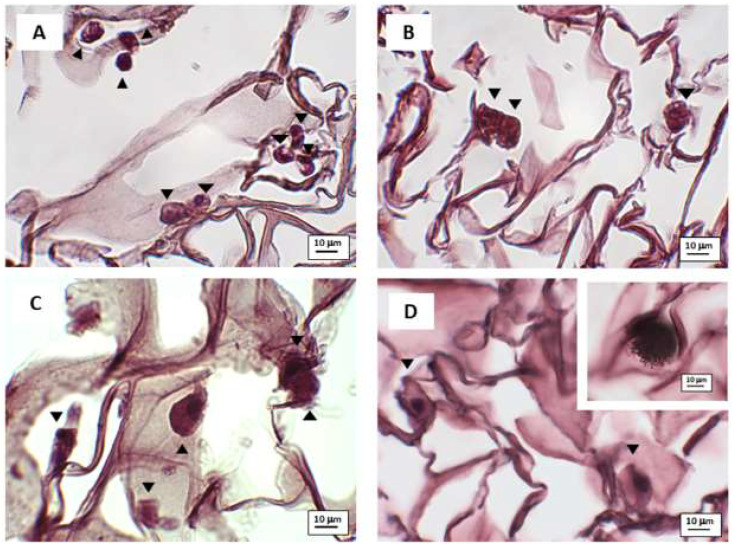
Photomicrographs of optical microscopy observation. hASCs were grown in FWM for 24 (**panel A**), 48 (**panel B**), 72 (**panel C**) and 96 h (**panel D**). The cells were stained with haematoxylin–eosin. The inset shows an apoptotic cell. Arrowhead indicates the cells.

**Figure 4 nanomaterials-10-01822-f004:**
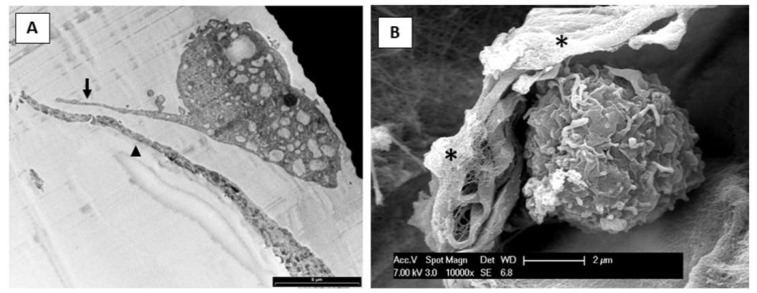
Photomicrographs acquired by TEM (**panel A**) and SEM (**panel B**) showing hASCs grown in FWM. In panel A, the arrow indicates the pseudopod-like structures while the arrowhead indicates the scaffold. In panel B a cell, covered by some collagen fibrils, appears to have colonized the scaffold. The asterisk indicates the scaffold. Scale bare in panel A is 5 μm.

**Figure 5 nanomaterials-10-01822-f005:**
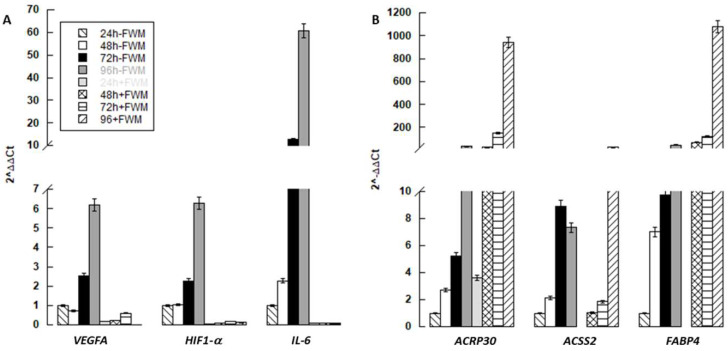
mRNA expression of markers related to angiogenesis (**A**) and differentiation (**B**) of hASCs grown in presence (+FWM) or not (−FWM) of FWM (*n* = 3).

**Figure 6 nanomaterials-10-01822-f006:**
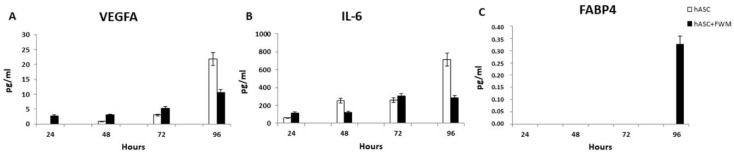
Histograms of enzyme-linked immunosorbent assay (ELISA) quantification of (**A**) VEGFA, (**B**) IL-6 and (**C**) FABP4 in the culture medium of hASCs grown in presence (+FWM) or not (−FWM) of FWM (*n* = 3).

**Table 1 nanomaterials-10-01822-t001:** Primers used in this work. FW = Forward primer; Rev = Reverse primer.

Gene Name	Primer Sequence 5′-3′	Melting Temperature (°C)	Accession Number
***Hs GAPDH***	*FW* CCCTTCATTGACCTCAACTAC	61.5	M17851.1
	*Rev* CATTGATCACAAGCTTCCCG	61.6	
***Hs β2m***	*FW* TTCTGGCCTGGAGGCTATC	60.0	AB021288.1
	*Rev* TCAGGAAATTTGACTTTCCATTC	59.0	
***Hs VEGFA***	*FW* GGAGTCCAACATCACCAT	60.5	AY047581.1
	*Rev* GCTGTAGGAAGCTCATCT	59.9	
***Hs IL-6***	*FW* ACTCACCTCTTCAGAACG	60.0	M14584.1
	*Rev* CCTCTTTGCTGCTTTCAC	60.4	
***Hs HIF1α***	*FW* CAAGTCCTCAAAGGACAG	59.7	AF304431.1
	*Rev* TGGTAGTGGTGGCATTAG	60.1	
***Hs FABP4***	*FW* AAGTCAAGAGCACCATAACCT	63.3	NM_001442.2
	*Rev* GCATTCCACCACCAGTTTATC	63.4	
***Hs ACRP30***	*FW* GGAAGGAGAGCGTAATGGA	62.7	NM_004797.3
	*Rev* AGTTGGTGTCATGGTAGAGAA	62.7	
***Hs ACSS2***	*FW* ATACAAGGTGACCAAGTTCTACA	63.3	NM_018677.3
	*Rev* GTGACAGGCTCATCTCCAA	63.3	

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
