# Peer review of "Effect of Nanostructured Scaffold on Human Adipose-Derived Stem Cells: Outcome of In Vitro Experiments"

_nanomaterials, 2020, doi:10.3390/nano10091822_

Round 1
Reviewer 1 Report
This manuscript reviews and provides some interesting data on the use of nanostructure scaffold on viability, differentiation and secretion of growth factors by hASCs with well known activity on tissue regeneration and angiogenesis.
The data is well presented and easy to follow. Unfortunately, it is limited by the lack of in vivo data to determine the efficacy and/or safety of this approach in pre-clinical animal models of tissue regeneration. This limitation reduces the impact of this manuscript.
The manuscript has some significant commercial information that may bias the information provided. This reviewer suggests that this commercial description is left in the Methods section and the information on the Integra® FWM-hASC mixture preparation is not supported by a photo and the patented name of his invention appears within a parenthesis in the description paragraph.
Author Response
REVIEWER_1
Comments and Suggestions for Authors
This manuscript reviews and provides some interesting data on the use of nanostructure scaffold on viability, differentiation and secretion of growth factors by hASCs with well known activity on tissue regeneration and angiogenesis.
The data is well presented and easy to follow. Unfortunately, it is limited by the lack of in vivo data to determine the efficacy and/or safety of this approach in pre-clinical animal models of tissue regeneration. This limitation reduces the impact of this manuscript.
The Reviewer is right and considering the importance of his comment, off the record, we enclose the preliminary data of our in vivo experiments that we intend to include in an incoming manuscript.
EXPERIMENTAL PLAN AND PRELIMINARY RESULTS OF in vivo EXPERIMENTS, conducted in our laboratory, on athymic CD1 nude mice grafted with four different conditions: 1- FWM alone; 2- FWM associated with hASCs; 3- FWM associated with hASC crude protein extract. 4- FWM associated with hASC-conditioned medium;
The manuscript has some significant commercial information that may bias the information provided. This reviewer suggests that this commercial description is left in the Methods section and the information on the Integra® FWM-hASC mixture preparation is not supported by a photo and the patented name of his invention appears within a parenthesis in the description paragraph.
Following the Reviewer suggestion, throughout the entire manuscript, we have deleted the commercial name of the scaffold. As the Figure 1 was taken in our laboratory and it is referred to the FWM-hASC mixture preparation, we think that it is useful to the reader’s comprehension

Reviewer 2 Report
The paper is well written and methodology is adequate
However, there is a lack of translational topics about applications of such scaffold.
In particular:
- the Authors should focus more on clinical transitional aspects of the nanostructured coatings. Indeed, they should include some studies about biomaterials, that can act as scaffold, and its clinical application for bone regeneration in dentistry (PMID: 22010090 ; PubMed ID: 22299085; DOI: 10.1902/jop.2010.090477; DOI: 10.1902/jop.2009.090156)
- Moreover, it’s important to emphasize that oral implants with such nanostructured scaffold could improve osseointegration in critical patients from the systemic point of view, in particular HIV positive patients that reported an increase in early failures of dental implants (PMID: 26238779; PMID: 25955953)
- Authors should discuss the potential role of hydroxyapatite as scaffold for osteogenic differentitation (PMID: 26753641) and potential applications for HiPS stem cells, that are not cited in the text (PubMed ID: 32188154)
- About further studies, Author should hypothesize more biomolecular analyses, as reported in papers about in vivo gene expression of osteoblasts in contact with bone substitutes and/or hydroxyapatite (PubMed ID: 21841997; DOI: 10.1902/jop.2009.080466)
- In the end, from a transitional/clinical point of view, they should emphasize that the aim of realizing innovative implant surface is always to avoid invasive techniques (PMID: 31140209) in favour of mini-invasive ones (PMID: 31781702)
Author Response
REVIEWER_2
Comments and Suggestions for Authors
The paper is well written and methodology is adequate
However, there is a lack of translational topics about applications of such scaffold.
We thanks the Reviewer for the suggestions. We revised the discussion in accordance with most of the advices. Unfortunately, some of the suggested papers are not available in open access and other are out of our research interest and knowledge, so we have not been able to cite them. We hope that this Reviewer is satisfied with the proposed revision.
In particular:
the Authors should focus more on clinical transitional aspects of the nanostructured coatings. Indeed, they should include some studies about biomaterials, that can act as scaffold, and its clinical application for bone regeneration in dentistry (PMID: 22010090; PubMed ID: 22299085; DOI: 10.1902/jop.2010.090477; DOI: 10.1902/jop.2009.090156)
See lines 274-278 of the revised manuscript and the new references 30, 31 and 32.
Moreover, it’s important to emphasize that oral implants with such nanostructured scaffold could improve osseointegration in critical patients from the systemic point of view, in particular HIV positive patients that reported an increase in early failures of dental implants (PMID: 26238779; PMID: 25955953)
This point is definitely out of our research interest and knowledge and would complicate the discussion of our results.
Authors should discuss the potential role of hydroxyapatite as scaffold for osteogenic differentitation (PMID: 26753641) and potential applications for HiPS stem cells, that are not cited in the text (PubMed ID: 32188154)
See lines 274-278 of the revised manuscript and the new references 30, 31 and 32.
About further studies, Author should hypothesize more biomolecular analyses, as reported in papers about in vivo gene expression of osteoblasts in contact with bone substitutes and/or hydroxyapatite (PubMed ID: 21841997; DOI: 10.1902/jop.2009.080466)
See lines 278-282 of the revised manuscript and the new reference 33.
In the end, from a transitional/clinical point of view, they should emphasize that the aim of realizing innovative implant surface is always to avoid invasive techniques (PMID: 31140209) in favour of mini-invasive ones (PMID: 31781702)
See lines 312 of the revised manuscript.

Reviewer 3 Report
In this manuscript, the authors showed Integra® Flowable Wound Matrix (FWM) reduced normal physiology of hASCs. This study has messages that could be referenced in clinical field. There are issues should be addressed before publication.
Line 17. In the word ‘ELISA’, already there is ‘assay’, so ‘assay’ should be deleted.
Line 19. In Line 293(discussion part), you suggested the possible scenario that can explain relationship between angiogenesis and hASC differentiation. But with this sentence alone, it is insufficient to support “Results suggested that, in presence of the scaffold, cell proliferation and the exocytosis of factors involved in the angiogenesis process, are reduced in favour of the expression of those genes involved in hASC differentiation.” So correct ‘in favour of’ to ‘and’
Line 21. grammar check.
Line 37. “nanoscopic scale” phrase may need to include specific size range.
Line 39. Is the expression ‘homeostasis regeneration’ proper thing?
Line 58. Almost same sentence with line 66. Need to paraphrase.
Line 93. There are no cytofluorimetric analysis data in the result.
Line 98-100. Why is there the difference between staminal/differentiation markers of cytofluorimetric and qPCR?
Line 115. typo: shacked should be shook
Line 174. ELISA assay can be changed to ELISA
Line 176. typo: additional h can be changed to additional hours
Line 188. “96 hrs culture” can be changed to “96 hours of incubation”
Figure 2B. y axis OD cell viability can be changed to OD or cell viability (OD)
Line 199. typo: insect should be inset
Figure 5A. typo: 96 + FWM can be changed to 96h + FWM
Figure 5 A, B. y axis 2^ddCt can be changed to "relative" mRNA "expression"
Line 227. ELISA assay can be changed to ELISA
Line 230. “96 hrs” can be changed to “96 hours”
Figure 6. Why isn’t the comparison data of HIF-a?
Figure 6B. There is no mark that indicates the significance of the data. Furthermore, the consistency of the data is absent.
Line 273. It could be better if you confirm the other tissue markers such as osteocyte, chondrocyte (the tissue that the hASC could be differentiated).
Line 273. “96 hrs” can be changed to “96 hours”
Line 297. grammar check.
Line 371. Ref 17 - DOI is missing.
Line 387. Ref 22 – DOI is repeated.
Line 392. Ref 24 – some part of DOI is missing.
- In the abstract, you referred about tissue regeneration and recovery of injuries. However, the data are only associated with angiogenesis. Thus, in the abstract, I propose you to revise the ‘tissue regeneration’ to ‘angiogenesis’.
- In the end of abstract, you proposed the use of conditioned culture medium. However, the numeric data about this suggestion is absent in the result. Therefore, It should be deleted in the abstract.
- Reference – Please match form when marking DOI
Example
- Ref 4. Doi: 10.1021/nn304966z
- Ref 5. https://doi.org/10.1080/23320885.2017.1407658 (“Doi” is not written)
- Ref 6. doi:10.1016/j.matbio.2016.06.002. (“d” is not capitalized)
Author Response
REVIEWER_3
Comments and Suggestions for Authors
In this manuscript, the authors showed Integra® Flowable Wound Matrix (FWM) reduced normal physiology of hASCs. This study has messages that could be referenced in clinical field. There are issues should be addressed before publication.
Line 17. In the word ‘ELISA’, already there is ‘assay’, so ‘assay’ should be deleted.
Revised as requested (see line 18 in the revised manuscript).
Line 19. In Line 293(discussion part), you suggested the possible scenario that can explain relationship between angiogenesis and hASC differentiation. But with this sentence alone, it is insufficient to support “Results suggested that, in presence of the scaffold, cell proliferation and the exocytosis of factors involved in the angiogenesis process, are reduced in favour of the expression of those genes involved in hASC differentiation.” So correct ‘in favour of’ to ‘and’
Following the Reviewer’s suggestion, we have modified the sentence as follow: “Results suggested that, in presence of the scaffold, cell proliferation and the exocytosis of factors involved in the angiogenesis process are reduced; differently, the expression of those genes involved in hASC differentiation appeared enhanced.” (see lines 20-21 in the revised manuscript).
Line 21. grammar check.
The Reviewer is right, the sentence is grammatically incorrect. We reframed as follow: “Furthermore, being the paracrine event the primary mechanism exerting the beneficial effects on injured tissues, these data propose the use of conditioned culture medium to replace cells.” (see lines 23-25 in the revised manuscript)
Line 37. “nanoscopic scale” phrase may need to include specific size range.
Revised as requested (see line 41 in the revised manuscript).
Line 39. Is the expression ‘homeostasis regeneration’ proper thing?
In the revised manuscript, we have substituted ‘homeostasis regeneration’ with “physiological regeneration” (see line 42 in the revised manuscript).
Line 58. Almost same sentence with line 66. Need to paraphrase.
Following the Reviewer’s suggestion, we have deleted the sentence “comprised of granulated cross-linked bovine tendon collagen and glycosaminoglycan” in the introduction (see lines 61-62 in the revised manuscript).
Line 93. There are no cytofluorimetric analysis data in the result.
The Reviewer is right. In the revised version of the manuscript, we now specified that “Cells were characterized by cytofluorimetric analysis, immunostaining, and qPCR; the methods and the results were described in Cherubino [16] and Borgese [17]” (see lines 94-95 in the revised manuscript).
Line 98-100. Why is there the difference between staminal/differentiation markers of cytofluorimetric and qPCR?
We first characterized hASCs by FACS analysis to be sure of the stemness of the cells used in the experiments. In a second time, we have completed the cell characterization investigating, by qPCR, the expression of genes involved both in stemness and in differentiation.
Line 115. typo: shacked should be shook.
Revised as requested (see line 118 in the revised manuscript).
Line 174. ELISA assay can be changed to ELISA.
Revised as requested (see line 177 in the revised manuscript).
Line 176. typo: additional h can be changed to additional hours.
Revised as requested (see line 179 in the revised manuscript).
Line 188. “96 hrs culture” can be changed to “96 hours of incubation”.
Revised as requested (see line 191 in the revised manuscript).
Figure 2B. y axis OD cell viability can be changed to OD or cell viability (OD).
Revised as requested (see new figure 2 in the revised manuscript).
Line 199. typo: insect should be inset.
Revised as requested (see line 204 in the revised manuscript).
Figure 5A. typo: 96 + FWM can be changed to 96h + FWM.
Revised as requested (see new figure 5 in the revised manuscript).
Figure 5 A, B. y axis 2^ddCt can be changed to "relative" mRNA "expression".
Revised as requested (see new figure 5 in the revised manuscript).
Line 227. ELISA assay can be changed to ELISA.
Revised as requested (see line 233 in the revised manuscript).
Line 230. “96 hrs” can be changed to “96 hours”.
Revised as requested (see line 236 in the revised manuscript).
Figure 6. Why isn’t the comparison data of HIF-a?
No justification for this observation except for the economic issue related to the purchase of the kit.
Figure 6B. There is no mark that indicates the significance of the data. Furthermore, the consistency of the data is absent.
As reported in lines 182-183 of the revised version, the results were expressed as mean+S.E. No Analysis of variance (ANOVA) was conducted for these data.
Line 273. It could be better if you confirm the other tissue markers such as osteocyte, chondrocyte (the tissue that the hASC could be differentiated)
As suggested, we have consolidated the discussion including osteocyte markers (see line 279-283 in the revised manuscript).
Line 273. “96 hrs” can be changed to “96 hours”.
Revised as requested (see line 285 in the revised manuscript).
Line 297. grammar check.
Checked as requested (see line 315 in the revised manuscript).
Line 371. Ref 17 - DOI is missing.
Corrected as requested
Line 387. Ref 22 – DOI is repeated.
Corrected as requested
Line 392. Ref 24 – some part of DOI is missing.
Corrected as requested
In the abstract, you referred about tissue regeneration and recovery of injuries. However, the data are only associated with angiogenesis. Thus, in the abstract, I propose you to revise the ‘tissue regeneration’ to ‘angiogenesis’.
The Reviewer is right. In the revised manuscript, we modified the sentence (see lines 14 in the revised manuscript).
In the end of abstract, you proposed the use of conditioned culture medium. However, the numeric data about this suggestion is absent in the result. Therefore, It should be deleted in the abstract.
We have accepted the Reviewer suggestion and properly corrected the sentence (see lines 23-25 in the revised manuscript).
Reference – Please match form when marking DOI.
Example
- Ref 4. Doi: 10.1021/nn304966z
- Ref 5. https://doi.org/10.1080/23320885.2017.1407658 (“Doi” is not written)
- Ref 6. doi:10.1016/j.matbio.2016.06.002. (“d” is not capitalized)
Corrected as reported in the journal guidelines.

Round 2
Reviewer 3 Report
I am satisfied with the revised manuscript